# Rapid evolution of the primate larynx?

Daniel L. Bowling[ID][1,2☯]*, Jacob C. Dunn[ID][2,3,4☯], Jeroen B. Smaers[5,6], Maxime Garcia[ID][2,7], Asha Sato[8], Georg Hantke[9], Stephan Handschuh[10], Sabine Dengg[11], Max Kerney[ID][3], Andrew C. Kitchener[9], Michaela Gumpenberger[ID][11], W. Tecumseh Fitch[ID][2]*

1 Department of Psychiatry and Behavioral Sciences, Stanford University, Stanford, California, United States of America, 2 Department of Behavioral & Cognitive Biology, University of Vienna, Vienna, Austria, 3 Behavioural Ecology Research Group, Anglia Ruskin University, Cambridge, United Kingdom, 4 Biological Anthropology, Department of Archaeology, University of Cambridge, Cambridge, United Kingdom, 5 Department of Anthropology, Stony Brook University, Stony Brook, New York, United States of America, 6 Division of Anthropology, American Museum of Natural History, New York City, New York, United States of America, 7 Animal Behaviour, Department of Evolutionary Biology and Environmental Science, University of Zurich, Zurich, Switzerland, 8 Center for Language Evolution, University of Edinburgh, Edinburgh, United Kingdom, 9 Department of Natural Sciences, National Museums Scotland, Edinburgh, United Kingdom, 10 VetCore Facility for Research, University of Veterinary Medicine Vienna, Vienna, Austria, 11 Klinische Abteilung für Bildgebende Diagnostik, University of Veterinary Medicine Vienna, Vienna, Austria

☯ These authors contributed equally to this work.
* dbowling@stanford.edu (DLB); tecumseh.fitch@univie.ac.at (WTF)

## Abstract

Tissue vibrations in the larynx produce most sounds that comprise vocal communication in mammals. Larynx morphology is thus predicted to be a key target for selection, particularly in species with highly developed vocal communication systems. Here, we present a novel database of digitally modeled scanned larynges from 55 different mammalian species, representing a wide range of body sizes in the primate and carnivoran orders. Using phylogenetic comparative methods, we demonstrate that the primate larynx has evolved more rapidly than the carnivoran larynx, resulting in a pattern of larger size and increased deviation from expected allometry with body size. These results imply fundamental differences between primates and carnivorans in the balance of selective forces that constrain larynx size and highlight an evolutionary flexibility in primates that may help explain why we have developed complex and diverse uses of the vocal organ for communication.

## Introduction

Recent years have witnessed a renaissance in research into the evolutionary and mechanistic basis of mammalian vocal communication. Two factors underlie this progress. The first is an extension of fundamental theoretical principles, initially developed for human speech, to a much broader range of vertebrate taxa, rendering it clear that source-filter theory and myoelastic-aerodynamic theory apply to a wide range of terrestrial species [1–4]. The second is the comparative application of these theories to large quantities of interspecific data, using phylogenetically controlled methods designed to address fundamental questions about trait evolution, including, e.g., the roles of sexual selection, "honest" indicators of caller characteristics, ecological factors, and morphological and neural specializations for vocal signals [5–10].

**Data Availability Statement:** All of the data used in this paper are recorded in the supporting information file S1 Data.xlsx.

**Funding:** This work was funded in part by Austrian Science Fund (FWF) DK Grant "Cognition & Communication" (#W1262-B29). DLB was

supported by a Lise Meitner grant from the FWF (#M1773-B24). JCD was supported by a grant from the Royal Society (RSG\R1\180340) and the Rhinology and Laryngology Research Fund. National Museums Scotland thanks the Negaunee Foundation for their generous support of a preparator who dissected the larynges use in the study. The funders had no role in study design, data collection and analysis, decision to publish, or preparation of the manuscript.

**Competing interests:** The authors have declared that no competing interests exist.

**Abbreviations:** CT, computed tomography; CV, coefficient of variation; F0, fundamental frequency; OU, Ornstein-Uhlenbeck; pANCOVA, phylogenetic ANCOVA; PC1, first principal component; pGLS, phylogenetic generalized least squares; RA, relative age at death; SSD, sexual size dimorphism; SS, specimen sex.

Much of this recent progress has focused on primates, in which acoustic investigations have been intense and a broad range of species has been studied [5,6,9,11]. At the same time, analyses of similar scope, but focused on brain size, have taught us that principles that apply generally within a clade like primates do not necessarily apply to other clades like carnivorans or ungulates [12]. Carnivorans in particular are of interest for comparative analyses focused on vocal communication because they are comparable to primates in terms of the range of habitats they occupy (e.g., terrestrial to arboreal), the social systems they exhibit (e.g., solitary to gregarious), and the body sizes they display ($10^{-2}$x to $10^{2}$x kg) [13,14]. Vocalizations in some carnivoran species have been studied in detail [15,16], and some comparative work has been done [17,18], but large-scale interspecific analyses applied across the carnivoran order have not been initiated.

Here, we compare a key mechanistic determinant of vocalization across primates and carnivorans. Our sample includes 55 species, ranging in size over three orders of magnitude in both clades: from pygmy marmoset (*Cebuella pygmaea*, mean body mass approximately 110 g) to Western gorilla (*Gorilla*, approximately 120 kg) in primates ($n = 26$); and from dwarf mongoose (*Helogale parvula*, approximately 280 g) to tiger (*Panthera tigris*, approximately 180 kg) in carnivorans ($n = 29$). We focus on the larynx (i.e., the main organ of vocal production), combining three-dimensional computer models built from X-ray computed tomography (CT) scans with detailed digital measurements, using a protocol designed to characterize gross features of laryngeal morphology consistently across species. Phylogenetically controlled comparisons of these data with specimen-specific body lengths reveal marked differences in the evolutionary trajectories of overall larynx size in the different clades. Parallel comparisons with acoustic vocalization data provide initial evidence that these differences are relevant to vocal communication.

## Results

Each larynx was characterized by a set of 10 measurements (Table 1 and Fig 1). Across species, the most variable measurements were associated with the ventral extent of the larynx, followed by our proxy for vocal fold length, and the distance between dorsal cricoid and ventral thyroid cartilages in the midsagittal plane. By contrast, the least-variable measurements were associated with the width of the thyroid cartilage in the coronal plane, followed by the diameter of

**Table 1. Laryngeal measurements sorted from most to least variable by CVs calculated on raw values.**

| Measurement name | Landmark numbers used | CV | Standardized loading on PC1 "larynx size" | Correlation with "larynx size" |
|---|---|---|---|---|
| Ventral cricoid height | 1 to 2 | 0.858 | 0.885 | 0.867 |
| Larynx height | 1 to 14 | 0.837 | 0.980 | 0.979 |
| Ventral thyroid height | 7 to 8 | 0.825 | 0.855 | 0.834 |
| Vocal fold length | [(9 to 12) + (9 to 13)]/2 | 0.717 | 0.960 | 0.944 |
| Crico-thyroid distance | 4 to 9 | 0.669 | 0.990 | 0.964 |
| Dorsal cricoid height | 3 to 4 | 0.648 | 0.940 | 0.887 |
| Apical cricoid depth | 2 to 4 | 0.587 | 0.990 | 0.973 |
| Basal cricoid depth | 1 to 3 | 0.585 | 0.975 | 0.943 |
| Basal cricoid width | 5 to 6 | 0.571 | 0.975 | 0.903 |
| Dorsal thyroid width | 10 to 11 | 0.559 | 0.975 | 0.920 |

Correlations are Pearson *r* values calculated between larynx size and the log-transformed measurements (all significant at $P < 0.0001$). Landmark numbers correspond to those in Fig 1.

Abbreviations: CV, coefficient of variation; PC1, first principal component

*Nomascus leucogenys* (Northern white-cheeked gibbon)

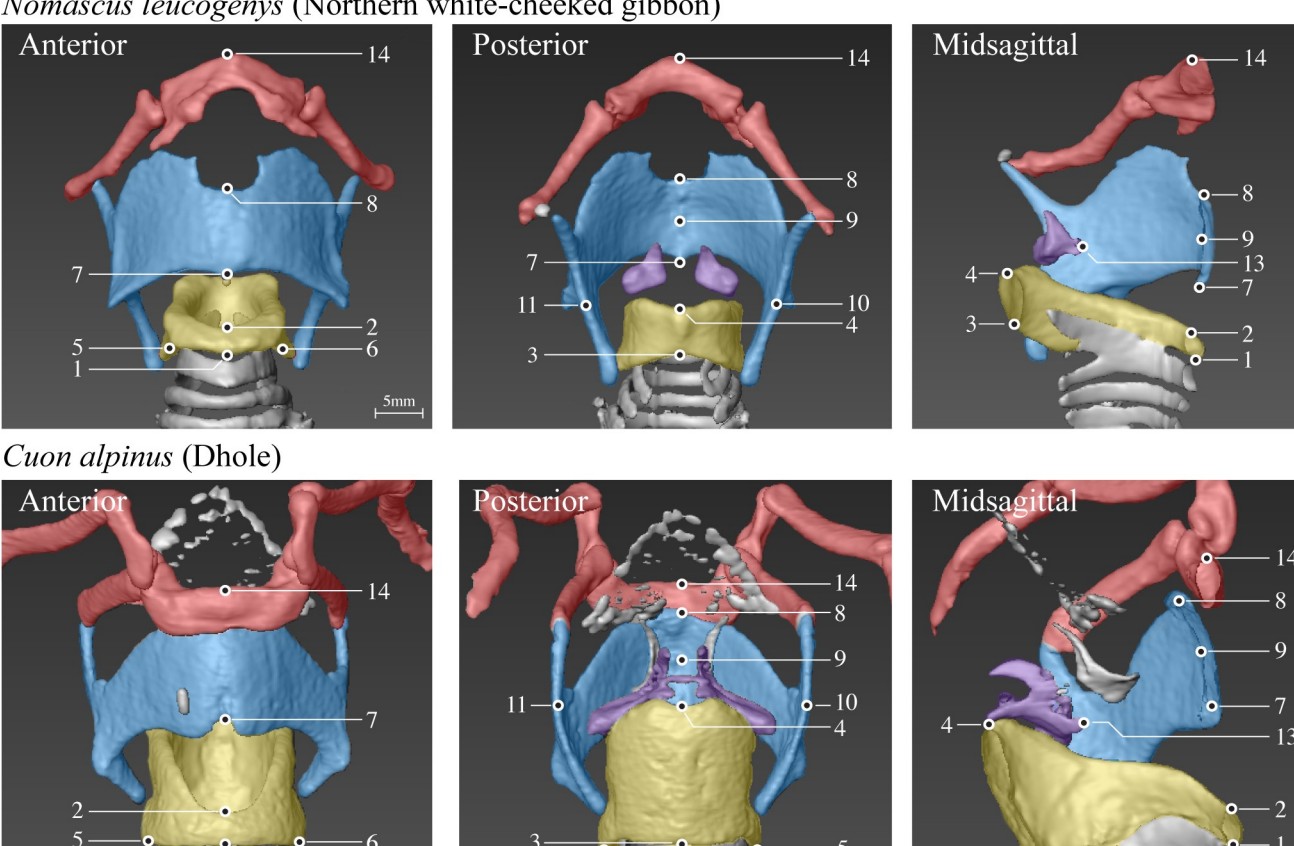

*Cuon alpinus* (Dhole)

**Fig 1. Laryngeal landmarks.** Three-dimensional digital larynx models from our data set showing a primate larynx (top row) and a carnivoran larynx (bottom row), each from three perspectives. The cricoid cartilage is shown in yellow, the thyroid cartilage in blue, the arytenoid cartilages in purple, and the hyoid bone in red (gray epiglottal and tracheal cartilages were not assessed). Midsagittal perspectives (right panels) show the left halves of each larynx, with the ventral aspects pointing right. The 14 anatomical landmarks placed on each larynx model (see Methods) are depicted as black circles. Note that landmark 12 is not depicted here, but located on the ventral tip of the right arytenoid vocal process in a location corresponding to landmark 13, but on the right side. Table 1 describes the 10 measurements derived from these landmarks.

the cricoid cartilage in the horizontal plane. For allometric comparisons, the dimensionality of the laryngeal measurement data was reduced by applying principal component analysis. The first principal component accounted for 91% of the variation in measurements across species and received high loadings from all 10 measurements (mean = 0.95, SD = 0.05). We refer to this first principal component as "larynx size" hereafter.

Fig 2 shows larynx size plotted against log10 body length for all 55 species. Allometric scaling was clear in both orders. Comparison of the primate and carnivoran regression lines, using phylogenetic ANCOVA (pANCOVA [19,20]), revealed similar slopes ($\beta_{prim}$ = 4.71 versus $\beta_{carn}$ = 4.73, $F_{3,54}$ = 2.162, $P$ = 0.148) but significantly different intercepts ($\alpha_{prim}$ = −8.04 versus $\alpha_{carn}$ = −9.09, $F_{3,54}$ = 8.177, $P < 0.001$). This indicates an evolutionary grade shift between primates and carnivorans, with primates exhibiting larger larynges for the same body length across our study sample. We estimated the magnitude of this difference by calculating the mean difference between our laryngeal measurements for eight primate-carnivoran species pairs with similar body length (specimens differed by <1 cm on average; see Fig 2). The results suggest that, for similar body lengths, the primate larynx is approximately 1.38× larger than the carnivoran larynx on average (SD = 0.38, range = 0.92–1.98). Importantly, we also found that residual

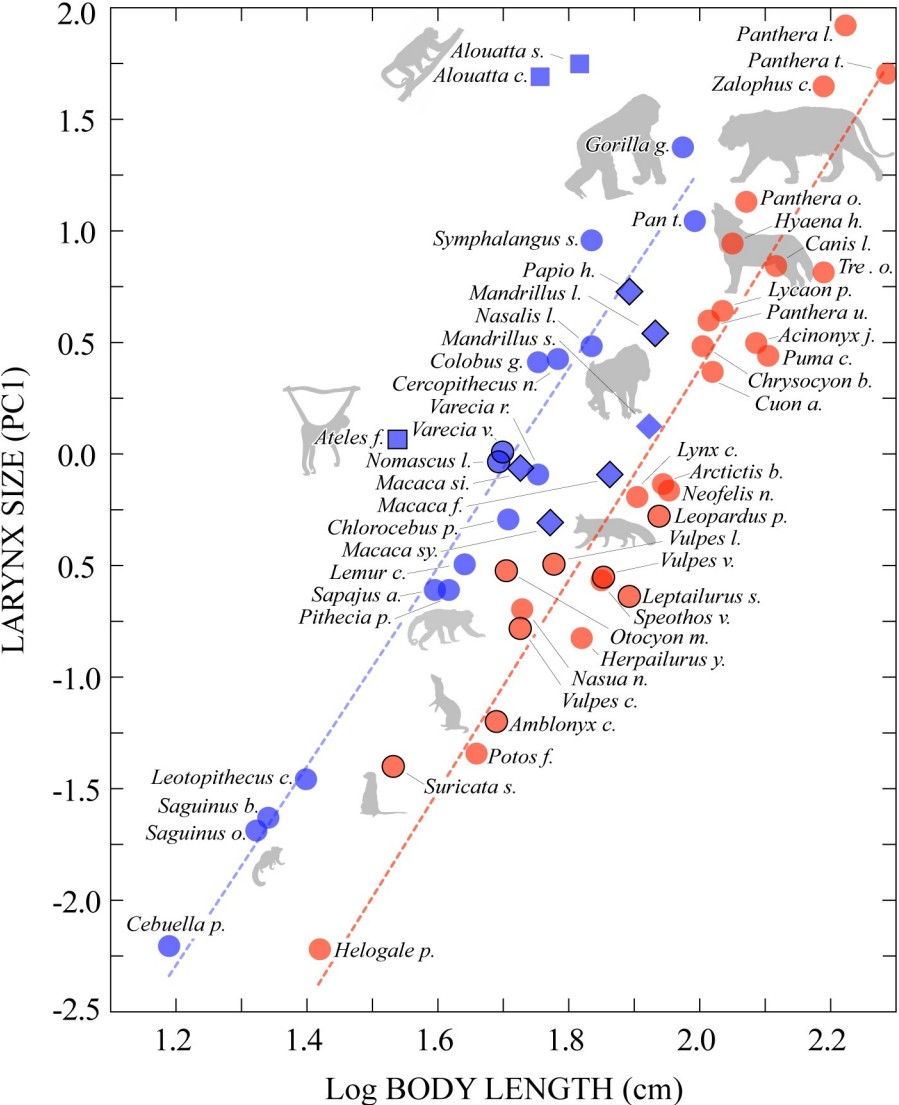

**Fig 2. Body length versus larynx size.** Base 10 logarithm of body length plotted against the PC1 of the larynx measurements for our sample of 26 primate (blue) and 29 carnivoran (red) specimens. Squares depict atelids and diamonds depict papionines, clades that stand out for having exceptionally large or small larynges among primates, respectively (see text). Specimens belonging to one of the eight pairs used to estimate the magnitude of the grade shift between primates and carnivorans are outlined in black. Regression lines display a significant grade shift between primates and carnivorans, as quantified by pANCOVA (see text). See Fig 3 for full species names corresponding to the abbreviated names shown here. The data used to create this figure are located in S1 Data, sheet B, columns B and C. pANCOVA, phylogenetic ANCOVA; PC1, first principal component.

variation of the primate allometry was significantly higher than residual variation of the carnivoran allometry (i.e., carnivorans exhibited stronger allometric integration; factor = 2.17x, $P$ = 0.03 obtained using a permutation analysis of primate-to-carnivoran rate ratios; see Methods), suggesting that the relationship of larynx size to body length is more flexible in primates.

To ensure that these results are not confounded by systematic differences between primates and carnivorans in the sex and age of our individual larynx specimens, or differences in the prevalence of species' sexual dimorphism between clades, we conducted additional phylogenetic generalized least squares (pGLS) regressions predicting larynx size as a function of body

length and the following covariates, each computed across as many of our specimens as possible: specimen sex (SS; M or F, $N = 52$), relative age at death (RA; specimen age at death/species' maximum life span, $N = 50$), and species' sexual size dimorphism (SSD; log10 average male mass/log10 average female mass; $N = 55$). Each model included one covariate as well as its interaction with body length. No significant effects of the covariates or their interactions with body length were observed (SS: $t_{41,37} = 1.597$, $P = 0.1188$; SS$_x$body length: $t_{41,37} = -1.103$, $P = 0.277$; RA: $t_{43,39} = 0.078$, $P = 0.938$; RA$_x$body length: $t_{43,39} = -0.108$, $P = 0.914$; SSD: $t_{43,39} = -0.069$, $P = 0.945$; SSD$_x$body length: $t_{43,39} = 0.039$, $P = 0.969$), indicating that the differences between primates and carnivorans shown in Fig 2 are robust to differences in the age, sex, and sexual dimorphism of the specimens and species we sampled. We also conducted a pGLS regression predicting larynx size as a function of body length with SS$_x$SSD as a covariate, to assess a possible effect of SS dependent on species' sexual dimorphism. The effect of the SS$_x$SSD covariate was not significant ($t_{41,37} = 1.460$, $P = 0.153$).

To further explore allometric patterns that may help explain the relative flexibility of the primate larynx to body-size relationship, we used multiregime Ornstein-Uhlenbeck (OU) evolutionary modeling to estimate whether additional grade shifts have occurred. These analyses reveal two additional grade shifts in primates (in Atelidae and Papionini) but no additional grade shifts in carnivorans. These results demonstrate that the allometric pattern between larynx size and body length that best fits our data includes a total of three grade shifts, with the additional shifts occurring in primates. Fig 3A displays the three grade shifts, their estimated phylogenetic locations, and the residual larynx size associated with each specimen. The signal-to-noise ratio ($\sqrt{\eta\phi}$) associated with this model was 7.29, demonstrating high effect size and thus high statistical power. The significance of the model fit to the data was confirmed by pANCOVA ($F_{4,54} = 8.1769$, $P < 0.001$). The first grade shift was toward larger residual larynx size in primates, estimated to have occurred at the root of their divergence from carnivorans. The second grade shift was toward even larger larynges in atelid primates (represented here by *Alouatta sara*, *Alouatta caraya*, and *Ateles fusciceps*), estimated at the root of their divergence from cebids. The third grade shift was toward smaller larynges in papionine primates (represented here by *Macaca sylvanus*, *Macaca fuscata*, *Macaca silenus*, *Papio hamadryas*, *Mandrillus sphinx*, and *Mandrillus leucophaeus*) at the root of their divergence from cercopithecines. Fig 3B illustrates the difference in larynx size, indicated by the first grade shift in situ between a similarly sized primate and carnivoran specimen.

Intuitive insight into the evolutionary diversification of the larynx in our sample can be gained by working backwards from the residual larynx sizes observed at the branch tips of our phylogenic tree to estimates of residual larynx size for its internal nodes. Accordingly, the ancestral phenogram in Fig 4A shows the evolutionary diversification of residual larynx size derived using a multiple-variance Brownian motion model that allows the rate of evolution along different lineages in a phylogeny to vary [21,22]. This depiction of evolutionary trait space shows that in order to achieve the variation observed in residual larynx size, the larynges of multiple primate lineages underwent rapid diversification. Fig 4B visualizes this difference in evolutionary rates using a standard Brownian motion Markov chain Monte Carlo procedure to estimate distributions of evolutionary rate that match our residual larynx size results and the associated phylogeny [23,24]. The resulting primate and carnivoran rate distributions demonstrate a significant difference in the amount of trait variance accumulated per unit time (rate ratio = 2.17x, $P = 0.03$).

To test whether these results are confounded by using body length as our proxy of body size —e.g., perhaps carnivorans are simply longer-bodied than primates—we repeated the analyses described above using body mass, rather than body length, as our proxy for body size. The results were very similar: primate larynges remained larger and more variable than carnivoran

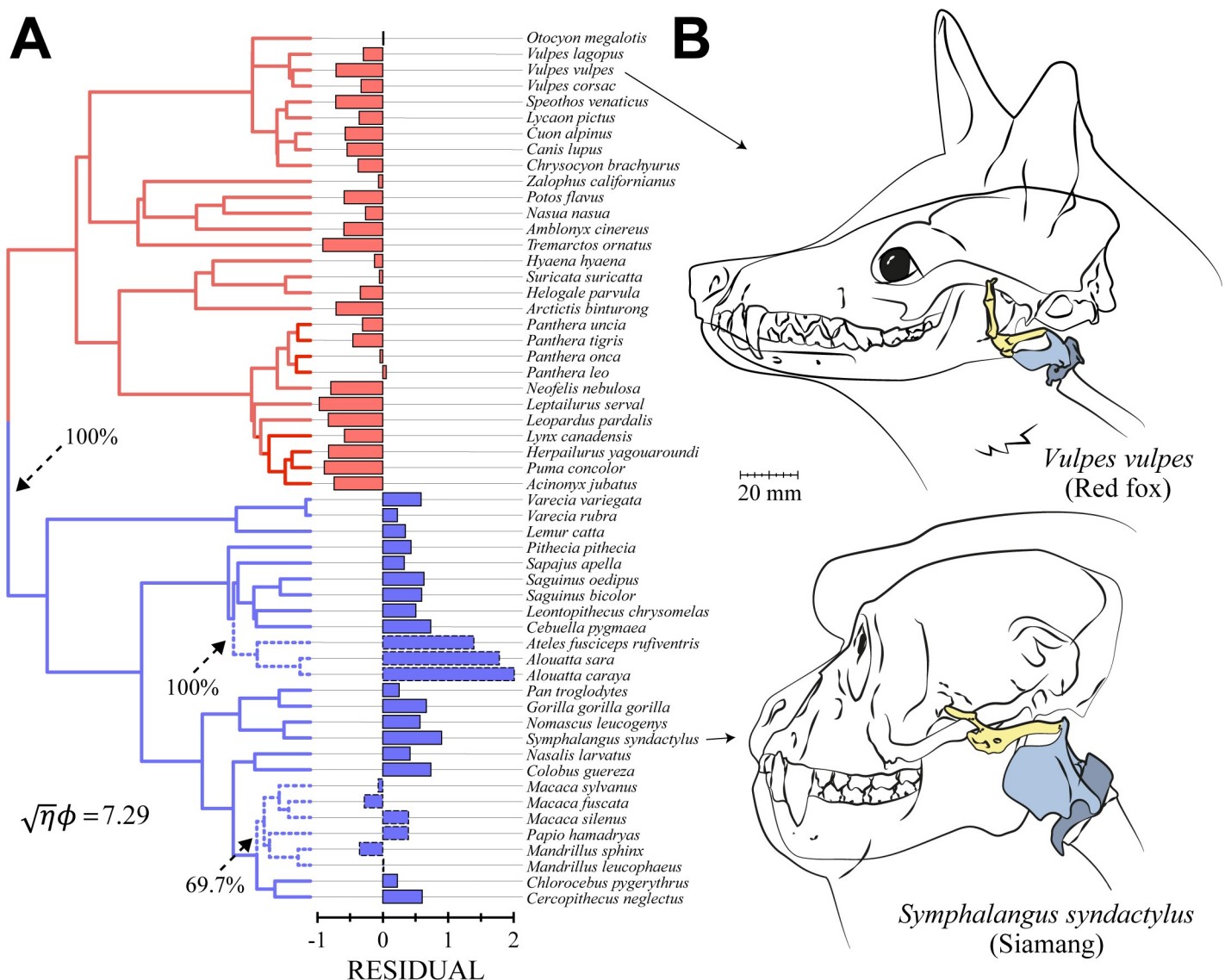

**Fig 3. Ornstein-Uhlenbeck model results.** (A) Phylogenetic tree and residuals from a pGLS regression of larynx size to log body-length. Carnivorans (red) exhibited smaller larynges than expected based on body size, whereas primates (blue) exhibited larger larynges. Among primates, atelids exhibited exceptionally large larynges (upper set of dashed lines), and papionines exhibited exceptionally small larynges (lower dashed lines). Arrows indicate where grade shifts in mean larynx size are estimated to have arisen; percentages indicate support for these estimations from a bootstrap analysis (see Methods). (B) Computer larynx models derived from CT scans depicted in situ for two species with comparable body lengths (71.4 cm for the red fox and 68.5 cm for the siamang), showing the larger relative size of the primate larynx. The data used to create this figure are located in S1 Data, sheet B, columns B and C. CT, computed tomography; pGLS, phylogenetic generalized least squares.

larynges, covariate effects and their interactions with body mass were not significant, and the average rate of primate laryngeal evolution was significantly greater than the rate of carnivoran laryngeal evolution. The only substantive difference was that in the mass analyses, only one grade shift beyond that between primates and carnivorans was observed, toward larger larynges in *Alouatta* at the root of their divergence from *Ateles*. Together, the mass analyses indicate that our findings are not an artifact of using body length as a proxy for body size. See S1 Text for full reporting of the body mass results.

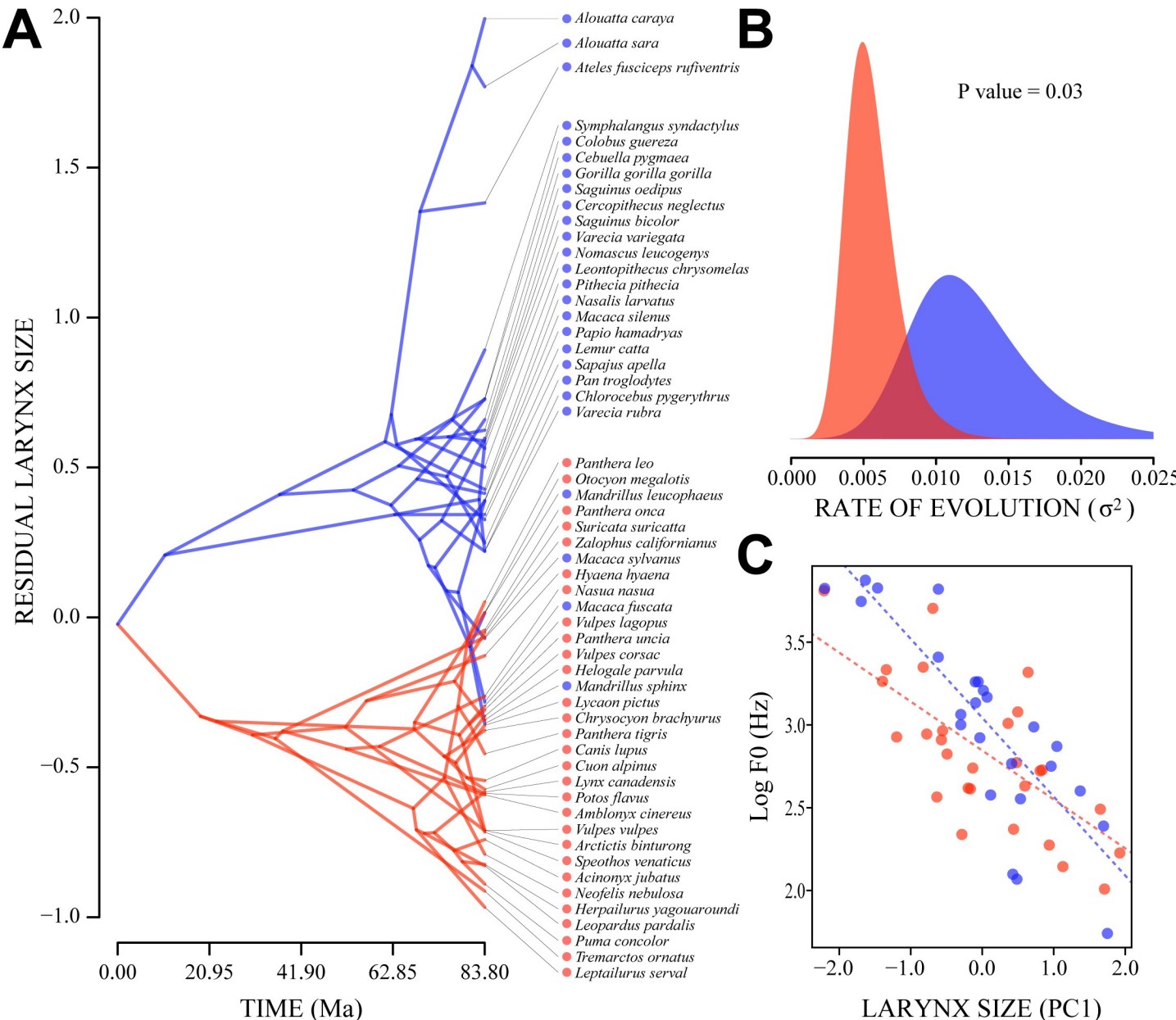

**Fig 4. Evolutionary rate analyses.** (A) Ancestral phenogram depicting divergence times plotted against residual larynx size from a pGLS regression of larynx size to log body length. Primate lineages (blue) are characterized by larger and more variable values than those of carnivoran lineages (red), as well as faster rates of change. (B) Probability density plots indicating the distributions of evolutionary rates required to produce the variance in residual larynx size observed in primates (blue) and carnivorans (red). The *P* value is the result of a permutation analysis of primate-to-carnivoran rate ratios (see Methods) and affirms the hypothesis that the two distributions are significantly different. (C) A comparison of larynx size and log mean-F0 in species-typical vocalizations (data from [9]). See S1 Fig for a labeled version of C. The data used to create Fig 4A and 4B are located in S1 Data, sheet B, columns B and C. The data used to create Fig 4C are located in S1 Data, sheet B, columns C and K. F0, fundamental frequency; PC1, first principal component; pGLS, phylogenetic generalized least squares.

Lastly, we compared larynx size with an acoustic feature measured directly from species-typical vocalizations across our sample to assess the potential link between our morphological results and vocal communication (Fig 4C). Fundamental frequency (F0) was selected as the most relevant acoustic feature here because of its important role in vocal communication and its close association with vocal fold length [1,25], which is strongly correlated with larynx size

in our sample ($r$ = 0.94; Table 1). Mean F0 data were obtained for 53 of our 55 species from [9]; *Lemur catta* and *Otocyon megalotis* were excluded because of insufficient data (see Methods). PGLS regressions between larynx size and mean F0 showed a significant negative relationship for primates ($t_{25,23}$ = −7.135, $P$ < 0.001) and for carnivorans ($t_{28,26}$ = −4.891, $P$ < 0.001). The slope of the primate regression was marginally steeper than that of the carnivoran regression ($\beta_{prim}$ = −0.476 versus $\beta_{carn}$ = −0.296, pANCOVA $F_{4,3}$ = 4.020, $P$ = 0.05), suggesting that a given change in larynx size is associated with a relatively larger change in mean F0 among primates. These results provide preliminary evidence that the phylogenetic patterns we observe in vocal morphology are related to the acoustics of vocal communication. The data used in all analyses described above are included in S1 Data.

## Discussion

The results described above highlight a clear pattern: relative to carnivoran larynges, primate larynges are significantly larger with respect to body size, more variable in size, and have evolved faster. Furthermore, this pattern is related to acoustic variation in the mean F0 of species-typical vocalizations, suggesting its relevance to vocal communication.

With respect to evolution, an important first point is that the phylogenetic variation in relative larynx size that we observe does not necessarily reflect variation in selective forces acting directly on larynx size. Logically, our results are consistent with variation in selection on larynx size, but also body size, or some combination of both. A parallel example comes from studies of relative brain size, where interspecific differences are commonly interpreted as reflecting selection for structural/functional enhancements of the brain, but large-scale comparisons akin to those performed here suggest a primary role for selection on body size [26]. Here, however, the evidence supports evolutionary change in larynx size rather than body size as the primary driver of the differences we observe: independent analyses of evolutionary rate applied to body length and "crico-thyroid distance"—a proxy for larynx size (see Table 1) measured in the same units as body length (mm) and thus suitable for direct comparison—indicate that larynx size has changed significantly faster than body size over time (by 2.15× for primates, 1.67× for carnivorans, and 1.79× overall; $P$ < 0.01 for all tests). This makes it unlikely that the differences in relative larynx size that we observe have been driven only by evolutionary changes in body size. Notably, a reversal of this general pattern was found for papionine primates, such that the rate of body-length evolution outpaced that of crico-thyroid distance by 1.19× (sample size [$n$ = 6] too small to appropriately test significance). Papionines also stand out here for having some of the smallest larynges relative to body length that we observed among primates, suggesting that understanding the constraints on laryngeal evolution in this clade may be particularly informative.

As demonstrated by the greater size variance and faster evolutionary rates observed in primates, larynx size is less tightly coupled to body size in this order than it is in carnivorans. This pattern of weaker allometric integration suggests that the primate larynx has been relatively more likely to respond to fluctuations in evolutionary pressure. That is, alterations in the balance of selective forces that maintain relative larynx size can be expected to have driven greater diversity in primates than carnivorans, which, as indicated by our comparisons with F0, would be expected to have had consequences for vocalization. In the following paragraphs we consider the hypothetical selective forces that act on larynx size and discuss whether or not differences in these forces between primates and carnivorans can plausibly account for our results.

Major adaptive hypotheses can be assigned to each major function of the larynx: protecting the airway during feeding, regulating the supply of air to the lungs, and vocal communication. A further relevant distinction is whether the proposed evolutionary trajectories are based on

directional selective pressure or a relaxation of selective pressure(s). Either can increase phenotypic variability, and multiple pressures can interact to determine phylogenetic trends.

A first possibility is that differences in relative larynx size between primates and carnivorans reflect the role of the larynx in protecting the respiratory system during feeding [27]. Although the trachea is mainly occluded by the epiglottis (not examined here) during swallowing, the larynx protects against aspiration of food or liquid via reflex closure and coughing when the epiglottis is bypassed [28]. Accordingly, increased variation in primate larynx size may be partially explained by a relaxation of selective forces related to diet and associated feeding behaviors relative to carnivorans. For example, it may be that the threat of choking on large pieces of minimally-chewed meat leads to selection pressure for smaller larynges in carnivorans, and that this pressure is somewhat relaxed in primates, which eat more plant material and spend more time chewing as a group [29]. It should be noted, however, that even though relaxed selection against choking provides a logical explanation for the overall differences in larynx size we observe between primates and carnivorans, it is not clear how this could explain the greater variance in larynx size observed among primates. Although carnivoran diets are certainly more based on animal matter than primate diets on average, there is considerable variation within both clades, from obligate herbivory among carnivorans (giant pandas) to obligate carnivory among primates (tarsiers).

A second possibility is that differences in relative larynx size between primates and carnivorans reflect the role of the larynx in respiration, e.g., in regulating the amount of oxygen that can enter the lungs and intrapulmonary pressure [30]. For example, differences in locomotor behavior between primates and carnivorans may place different demands on oxygen metabolism [31] and the capacity to stiffen the thorax by increasing intrapulmonary pressure [28]. Both factors are closely related to substrate use, which although variable in both orders, is more often terrestrial for carnivorans and arboreal for primates [32,33]. Notable exceptions among the carnivorans studied here include *Potos flavus*, *Nasua nasua*, and *Arctictis binturong*, all of which are relatively arboreal [34]. If the larger larynges of primates observed here were hypothesized to reflect selection (directional or relaxed) related to respiratory function, no support is found among these comparatively arboreal carnivorans, which all possess the relatively small larynges determined to be typical of carnivorans here. At the same time, some support is found among the papionine primates, which are relatively terrestrial compared to other primates and fit the pattern of having relatively small larynges. Even so, it is not clear how a relatively small/large larynx would benefit the muscular and locomotor requirements of a terrestrial/arboreal lifestyle, respectively. A related alternative hypothesis is that increased oxygen supply supported by the larger primate larynx serves brain function.

A third possibility Is that differences in relative larynx size between primates and carnivorans reflect the role of the larynx in vocal communication. Here, multiple hypotheses have been proposed to explain interspecific differences, and we consider the relevance of our data to each in turn. The acoustic adaption hypothesis suggests that vocalizations are optimized to maximize their transmission in different habitats [35,36]. The majority of primate species are found in tropical forest environments, whereas carnivorans occupy a greater diversity of habitats. Forest environments diminish the value of visual signals, while providing relatively constant physical conditions for acoustic signal transmission [35]. Together with the fact that primates are relatively microsmatic, this difference in habitat may increase selective pressure to produce vocalizations with frequency content suited to propagation in a forest habitat, i.e., the lower frequencies made possible by larger larynges [35]. This hypothesis appears compatible with the larger relative size of the primate larynx, but not its increased size variation. If variation in larynx size is contingent upon variation in habitat, carnivorans would be predicted to exhibit greater larynx size variation, not less, as observed here.

The sexual selection hypothesis proposes that larger larynges and the lower-frequency vocalizations they afford have been selected for acoustic size exaggeration in deterring opponents during intrasexual competition and/or attracting mates during intersexual choice—as appears to be the case in howler monkeys [7]. More broadly, sexual dimorphism in larynx size (and voice F0) appears to be strongly influenced by mating system in anthropoid primates (including humans), with polygyny favoring increased dimorphism [37]. This suggests that sexual selection may be important to understanding the kind of systematic differences in larynx size we observe here. This would be particularly true if the average strength of sexual selection differed between our samples of primates and carnivorans and/or if these samples were discrepant in terms of the sexes of specimens included (they were not; see Methods). However, using SSD as a proxy for the strength of sexual selection, we failed to find a significant difference between our primates and carnivorans. Additionally, when added as a covariate, SSD did not account for significant variation in the allometries between larynx and body size. A final point here is that although there is no carnivoran analogue for the enlarged larynges of howler monkeys (that we are aware of), the next most-supported OU model showed a shift toward relatively larger larynges in the roaring cats (*Panthera*) compared to the purring cats. Intriguingly, pantherines also possess enlarged vocal pads and an extremely elastic stylohyoid ligament [38,39], which may serve to further exaggerate body size in vocalizations (often performed in the darkness of night). Thus, although sexual selection and acoustic size exaggeration may play roles in accounting for some of the within-order variation we observed, it appears unlikely to account for overall differences between primates and carnivorans.

Finally, the social complexity hypothesis proposes that the larger and more varied larynges of primates reflect selection for enhanced vocal communication, as might be required to maximize the benefits and minimize the costs of social living. Primates and carnivorans exhibit fundamental differences in social behavior [40], with primates being more likely to form larger, more stable groups characterized by strong bonds and more close-contact time. In our sample, median group size was 13.8 for primates and 2.6 for carnivorans (Mann-Whitney $U = 121$, $P < 0.0001$; see Methods). Primate group size also tends to be more variable [41]. In our sample, median absolute deviation in group size was 51.9 for primates and 32.6 for carnivorans. Our results may be consistent with this hypothesis insofar as the more "complex" social lives of primates can be expected to result in pressure for larger larynges, which could potentially provide a more robust framework matched to the increased mechanical forces expected from more frequent use [42]. However, preliminary comparisons of larynx size and social group size data failed to show any significant correlation between these variables in our sample, for primates (Spearman $r = 0.11$, $P = 0.6094$), carnivorans ($r = 0.02$, $P = 0.9374$), or all species considered together ($r = 0.15$, $P = 0.2762$). As a stark example, consider that howler monkeys have some of the largest larynges relative to body size of any known species and yet live in relatively small social groups compared to many other primates [6]; conversely, papionines have relatively small larynges but live in relatively large social groups. These data suggest that social complexity, at least as far as it can be approximated by simple measures of social group size (see [43] for discussion), is unlikely to explain the clade-wise differences in larynx size that we observe.

A central obstacle in choosing among these hypotheses is that our data only allow us to assess primates and carnivorans with respect to one another. As a result, we cannot be certain whether the primate larynx is relatively large and variable or the carnivoran larynx relatively small and constrained. One reason to expect the former is that only primates appear to include additional grade shifts, as evidenced, e.g., by laryngeal hypertrophy in howler monkeys. But settling this issue will ultimately require more context from additional comparisons, e.g., with Artiodactyla, Rodentia, and/or Chiroptera, in which hammer-headed bats provide another

known example of laryngeal hypertrophy [44]. Only such comparisons can tell us how the variability observed here ranks among other mammalian orders. In addition to taking several years to source and analyze often rare specimens, such comparisons will be complicated by considerable differences in life history and body size, largely circumvented here by selecting clades with significant overlap in both domains.

Future studies will also benefit from the inclusion of multiple specimens per species, as error from within species variation in larynx size was not controlled here [45]. However, with regard to the magnitude of this error and its potential to bias our results, we note that our samples cover over three orders of magnitude in body length. This means that interspecific variation was much larger than the intraspecific variation, a fact that can be expected to mitigate the influence of intraspecific error on our main results. Importantly, we also controlled for the age and sex of our individual specimens, as well as sexual dimorphism at the species level, buffering against error variance arising from associated individual and intraspecific differences. A final important limitation to consider here is our focus on cartilage and bone only. This was a consequence of using CT with unfixed and unstained specimens, which remain viable for functional phonation studies. Although the cartilaginous structure of the larynx clearly correlates with its acoustic capabilities (see Fig 4C), it is ultimately the soft tissues of the larynx that are responsible for vocal production. Accordingly, studying evolutionary patterns of soft tissue variation will be important, particularly if this work continues to explore the vocal consequences of evolved laryngeal morphology.

In conclusion, the work described here demonstrates clear differences in larynx size and larynx-size variation between primates and carnivorans, reflecting fundamental differences in underlying rates of evolutionary change. These differences are consistent with multiple selective hypotheses, including directional selection resulting from cladistic differences in ecology and social structure that impact vocal communication, as well as relaxed selection on the feeding and respiratory functions of the larynx in primates relative to carnivorans. Our results open an exciting new avenue of study focused on laryngeal variation among further mammalian clades, which will provide the context required to determine how particular the differences we observe here are to the evolution of the primate larynx. If the relative flexibility of the primate larynx is robust to future analyses with more clades, it would indicate an increased capacity to explore trait space in our lineage, which may in turn explain why primates have developed such diverse and complex uses of the vocal organ.

## Methods

### Ethics statement

This research was conducted in accordance with European Union Directive 2010/63/EU. Approval by a specific IACUC committee was not required. Cadavers were donated to National Museums Scotland by the zoos of origin, and our use of their laryngeal specimens here was approved by the museum on a case-by-case basis.

### Specimens

The specimens used in this study lived and died in zoos throughout northern Europe. Death was followed by a postmortem examination performed by local veterinary staff at the zoo of origin after death, and the cadavers (excluding the digestive system) were frozen at −20˚C. The cadavers were then shipped on ice to National Museums Scotland for processing and preservation. Larynges, from the tongue to below the larynx, were excised during specimen preservation and refrozen for shipment on ice to Vienna, Austria. Upon arrival in Vienna, specimens were unpacked, thawed, and cleaned with saline before being mounted and refrozen in

preparation for X-ray CT scanning. Any specimens judged to be in poor condition at this stage (e.g., because of desiccation, decomposition, or insult) were excluded from further consideration. Selected specimens were mounted on polystyrene foam plates with ventral aspects facing upwards, and toothpicks were inserted into the foam on either side to prevent lateral rolling. During mounting, care was taken to avoid distortion and to approximate in vivo posture to the degree possible under the circumstances. The selection of which specific specimens/species to include in our analysis was based on balancing six factors: (1) availability; (2) including a wide range of body sizes in both orders; (3) maximizing phylogenetic variation; (4) maximizing tissue quality; (5) balancing sex across specimens; and (6) body length representativeness. If multiple specimens were available for a species, the one closest to estimated species' mean was selected (see S2 Fig for a comparison of specimen body length and estimates of species-typical ranges).

## CT scanning

The majority of the selected larynx specimens ($n$ = 49) were large enough to be scanned using a clinical CT scanner (Siemens Emotion 16, Munich, DE). Plate-mounted specimens were placed directly on the scanner gurney, rolled into the scanner bore, and scanned frozen. Depending on specimen size, source voltage was either 110 or 130 kV and beam intensity was either 80 or 90 mA (70 or 130 mAs). Each reconstructed slice (maximum of 1,215) measured $512 \times 512$ pixels. Depending on sample size, resolution of reconstructed volumes was between 238 and 369 $\mu m^2$ in the XY direction with slice thickness either 200 or 500 μm. Scan time was approximately 1 minute per specimen. Owing to their size, the six smallest specimens (*Cebuella pygmaea*, *Saguinus oedipus*, *Leontopithecus chrysomelas*, *Saguinus bicolor*, *Helogale parvula*, and *Suricata suricatta*, see S2 Fig) were scanned with an Xradia microXCT-400 scanner (Carl Zeiss X-ray Microscopy, Pleasanton, CA) equipped with the 0.4× lens. Because of the longer scan times associated with this technique (approximately 3 hours per specimen), these specimens were thawed, remounted upright inside sealed 50-ml Falcon tubes packed with plastic drinking straws for support and filled to 5 ml with phosphate-buffered saline to prevent dehydration (no contact with specimen). During this remounting process, care was taken to avoid distortion and approximate in vivo posture to the degree possible. Immediately after remounting, these specimens were placed in a tube rack that held them upright in the scanner bore and scanned with source voltage = 40 keV and beam intensity = 200 μA. Projections were recorded with 1-second exposure time (camera binning = 4) and an angular increment of 0.25˚. Reconstructed slices measured $512 \times 512$ pixels. Depending on sample size, isotropic voxel resolution of reconstructed volumes varied between 29 and 72 $\mu m^3$.

## Larynx models and measurement

Computer models of each larynx were created from the CT data to facilitate detailed measurements of the laryngeal cartilages in a nondestructive and consistent manner across species. Models were constructed, using Amira 3D data visualization software (version 5.6.0; FEI, Hillsboro, OR), and a detailed step-by-step protocol divided into four parts. The first part described model construction: (1) importing the CT data into Amira; (2) gradually adjusting the lower limit of the X-ray absorption window to visualize only smooth surfaces of the laryngeal cartilages and hyoid bone; and (3) using this lower limit to threshold Amira's "isosurface" function, which creates three-dimensional surface models of visualized data. The second part described the establishment of a framework for landmark placement: (4) defining a midsagittal plane by manually rotating and translating an Amira "ObliqueSlice" object to divide the cricoid cartilage in half along its dorsal-ventral axis; and (5) defining a midcoronal plane by

extracting the parameters of the midsagittal plane (using Amira's "getPlane" function), entering them into a custom Matlab script (version 2016a; MathWorks, Nantik, MA) that generated new Amira parameters for a perpendicular coronal plane, using these new parameters to define a second ObliqueSlice (using Amira's "setPlane" function), and manually translating this second ObliqueSlice to divide the cricoid cartilage in half along its medial-lateral axis. The third part described the placement of anatomically defined landmarks using Amira's "Landmark editor" (see Fig 1): (6) at the intersection of the cricoid cartilage and the midsagittal plane, the cricoid's ventral basal extent (#1), ventral apical extent (#2), dorsal basal extent (#3), and dorsal apical extent (#4); (7) at the intersection of the cricoid and the midcoronal plane, the cricoid's basal right and left lateral extents (#5 and 6); (8) at the intersection of the thyroid cartilage and the midsagittal plane, the thyroid's ventral basal extent (#7), ventral apical extent (#8), and the midpoint between these (#9); (9) on the thyroid cartilage at the level of landmark #4, the thyroid's dorsal right and left lateral extents (#10 and 11); (10) on the arytenoid cartilages at the ventral extent of the right and left vocal processes (#12 and 13); and (11) at the intersection of the hyoid bone with the midsagittal plane, the hyoid's apical extent (#14). The fourth part of the protocol described measurement: (12) exporting the coordinates of landmarks #1 to #14 as a text file; and (13) using a second custom Matlab script to read this text file and calculate the 10 Euclidean distances between specific pairs of points used to characterize laryngeal morphology. These measurements are named and defined in Table 1. Importantly, the anatomical locations of the 14 landmarks were not determined until after an initial inspection of all 55 of the digital larynx models included here (i.e., steps 1 to 5 were completed for all specimens before steps 6 to 13 were initiated). This allowed us to select anatomical locations where landmarks could be reliably placed across all specimens.

Specimen body length was defined as the distance between the ischium of the pelvis and the tip of the snout in carnivorans, or the top of the skull in primates. Body lengths were obtained from the same individual from which the larynx was excised for all but *Nasalis larvatus* and *Herpailurus yagouaroundi*. Average body lengths from [46,47] were substituted for these specimens. Body length was preferred over body mass, because the latter was only available for a small subset of our specimens, owing to postmortem procedures at the zoos of origin, and because we considered length to be less affected by zoo diet than mass (analyses based on average body mass are provided in S1 Text nonetheless). Laryngeal and body length measures were log-transformed (base-10), because these variables change on a proportional scale, whereas our linear models quantify variation on a linear scale.

### Principal component analysis

After log-transformation, the laryngeal measurements (see S1 Data) were subjected to principal component analysis in R (version 3.2.3; The R Foundation, Vienna, Austria), using the "psych" package function "principal" [48] to determine the factors underlying laryngeal variation across species. Initial analyses, configured to extract 10 components (without rotation), revealed only a single component with an eigenvalue >1 for both orders. Following Kaiser's criterion [49], we thus reconfigured the principal component analyses to extract a single component (rotation = varimax). The results indicated that this single component—which we refer to as "larynx size"—accounted for the vast majority of variation in laryngeal measurements, receiving high loadings from all 10 measurements (see Table 1).

### Phylogenetic comparative methods

A measure of relative larynx size for use in the statistical analyses was derived by calculating residuals from a pGLS regression of larynx size to body length across all species. A lambda

model showed that phylogenetic signal was high for this measure (λ = 0.956), which thus validly represents larynx size after controlling for body size and phylogenetic nonindependence due to shared ancestry. PGLS regressions were also used to model the effects of covariates on the relationship between larynx size and body length. The trees used for all phylogenetic analyses were obtained from [50] for primates and [51] for carnivorans.

To estimate the phylogenetic location of evolutionary grade shifts in mean relative larynx size in our sample, we used multiregime OU evolutionary modeling. This approach estimates where shifts in trait values occur along the branches of a phylogeny directly from the data. OU models differ from more widely used Brownian-motion models in that OU models incorporate additional parameters to account for changes in mean trait value (θ) and the strength at which the population moves from one mean to another (α). Several such OU methods have been proposed using different parameter maximization algorithms. Here, we checked estimations between the two most recent methods, "l1ou" and "phylogenetic expectation-maximization" [52,53], and confirmed that both yielded the same results for the model presented in Fig 3. The effect size associated with this OU model was quantified using the signal-to-noise ratio ($\sqrt{\eta\phi}$) and by running a bootstrap analysis. The signal-to-noise ratio demonstrates high power when >1 [54]. Bootstrap analysis was run using the "l1ou" method [53] and further confirmed strong support for the estimated grade shifts.

To test whether the estimated OU model provided significant results, we used pANCOVA [19,20] as a least-squares approach. PANCOVA provides a confirmatory test, indicating whether the clade-wise differences in mean relative larynx size, estimated by the OU model, are statistically significant by comparing the intercepts of the associated larynx size to body length allometries. We also used pANCOVA to test for differences in fit between alternative OU models, extending it to ask, e.g., whether a 2-grade or a 3-grade OU model provided a better fit to the data. The results of the OU model presented in Fig 3 were thus validated using pANCOVA. Although OU modeling approaches provide a powerful way to estimate patterns of evolutionary diversification, they necessarily entail more uncertainty than confirmatory least-squares hypothesis-testing approaches. Whereas OU modeling approaches assume more statistical parameters in order to estimate changes along individual branches of a phylogeny, pANCOVA is focused on observed tips only and does not infer changes along individual branches of a phylogeny, providing greater statistical power for confirmatory testing. It is clear, however, that shifts in trait value among clades observed at tips are expected to be the result of evolutionary changes that arose deeper in the phylogeny. The results of pANCOVA using only observed tips are thus expected to align with the estimations based on OU modeling approaches.

To estimate trait values for the internal nodes of our phylogeny, we used an ancestral estimation approach. Although not appropriate for hypothesis testing, such approaches provide an intuitive visual depiction of the evolutionary diversification of a trait. Traditionally, a standard Brownian-motion model has been used for these purposes. Here, however, we use a multiple-variance Brownian-motion approach, because this better accommodates possible changes in the rate of evolution that occur along different lineages in a phylogeny [21,22]. Considering that relative larynx size was found to entail both shifts in mean and rate among clades, a multiple-variance Brownian-motion model is more appropriate here than a standard Brownian-motion model in which rate is fixed. The results of these analyses are displayed in Fig 4A.

To test for differences in rate of change in relative larynx size in primates versus carnivorans, as well as rate of change in larynx size versus body size, we tested whether the ratio of the Brownian-motion rate in primates relative to the rate in carnivorans is significantly higher than unity, using a permutation analysis [24,55,56]. If this rate ratio is indeed higher than

unity, the rate in primates can be considered to be significantly higher than in carnivorans. The Brownian-motion rate parameter is the most appropriate measure to perform this test, because the Brownian motion model is a pure rate model (i.e., it quantifies rate, not direction). The interpretation of rate of evolution (quantified as $\sigma^2$ in the standard Brownian-motion model) corresponds to the amount of variance accumulated per unit time. Clades with a higher rate accumulate more trait variance per unit time, resulting in more observable trait variance and weaker allometric integration. To visualize the rate difference between primates and carnivorans in residual larynx size, we depict the posterior distributions of Markov chain Monte Carlo estimates of the respective Brownian-motion rate parameters (results of the rate analysis are displayed in Fig 4B).

## Specimen age, sex, sexual dimorphism, and species social group size

The potentially confounding effects of specimen age at death, SS, and species sexual dimorphism were addressed by modeling these factors as covariates in pGLS regressions predicting larynx size as a function of body length (and mass; see S1 Text). A separate regression was considered for each covariate and its interaction with body length. Descriptions of how the data for each covariate were sourced follow below. All data are provided in S1 Data.

Previous research in humans and nonhuman mammals has reported age effects on laryngeal geometry [57,58], indicating that the ages of the animals whose larynges were studied here may be a determining factor in their morphology. This could confound our results if there were a systematic age bias between our primates and carnivoran samples. To assess this possibility, we obtained age at death data for 50 of our specimens from National Museums Scotland's research records (missing: *Vulpes lagopus*, *Lynx canadensis*, *Potos flavus*, *Puma concolor*, and *Vulpes vulpes*). To account for differences in longevity across species (e.g., a chimpanzee might live 50 years in captivity, whereas a bush dog might only live 15), we normalized these data by dividing each specimen's age at death by maximum recorded longevity for its species (as recorded in [59]) to calculate "relative age." Maximum longevity was preferred over mean for normalization, because the latter was often distorted by exceptionally early deaths in [59], especially when sample size was small. A further way in which the age at death of our specimens could have confounded our results is through interaction with sexual maturity. For example, if the primate specimens studied here tended to be sexually mature at death and the carnivorans not, this could explain their systematically larger larynges. To assess this possibility, estimates of age at sexual maturity were obtained for as many of our study species as possible from a published compilation of zoo records ($N = 32$) [60] and compared with our data for age-at-death. For all but one of these 32 cases, the specimen we selected to represent its species came from an animal that was sexually mature at death (exception: *Vulpes corsac*; age at death = 6 months, estimated age at sexual maturity = 10 months). Although we could not determine age at death and sexual maturity for the remaining 23 of our specimens, taking these data as representative suggests that the majority are likely to have been sexually mature at death.

The sex of our specimens represents another potential confounding factor for our results. For example, if males tend to have larger larynges than females, and more males were included in our sample of primates than of carnivorans, then the differences we observed between these clades could be due to SS rather than a true difference between primates and carnivorans. However, sex data for our specimens from National Museums Scotland's research records indicate that our sample was closely matched, including 15 females, 10 males, and 1 unknown (*Chlorocebus pygerythrus*) for primates and 16 females, 11 males, and 2 unknowns

(*Herpailurus yagouaroundi* and *Tremarctos ornatus*) for carnivorans. SS (male or female) was nonetheless included as a covariate in a pGLS model predicting larynx size as a function of body length as a further point of caution. Another way in which sex could have impacted our results is through potential differences in sexual dimorphism between primates and carnivorans—for example, if we happened to select male specimens from highly dimorphic species for primates but females from highly dimorphic species for carnivorans. In this way, SS could have impacted our results even though male/female ratios were approximately the same in our sample of primates and carnivorans. To examine this possibility, we calculated an index of SSD for each species (log10 average male mass/log10 average female mass). Comparison of the SSD index between primates and carnivorans did not indicate a significant difference in SSD strength between orders. Nevertheless, we also included SSD as a covariate in a pGLS model predicting larynx size as a function of body length. Finally, we tested whether there was an effect of SS that was dependent on SSD by including the interaction between SSD and SS as a covariate in a pGLS model predicting larynx size as a function of body length.

Data describing social group size were obtained for all 55 of the species considered here. Social-group-size data for primates were mostly obtained from a single source [46], which typically reported minima and maxima for each species, allowing us to calculate an average represented by the midpoint of these ranges. For *Gorilla gorilla* and *Pan troglodytes*, mean group size was explicitly stated in [46] and used here instead of the midpoint. Social-group-size data for *Ateles fusciceps*, *Mandrillus sphinx*, and *Papio hamadryas* were not included in [46] and were instead obtained from [61] (*A. fusciceps*), [62] (*M. sphinx*), and [63] (*P. hamadryas*). Social-group-size data for carnivorans were mostly obtained from a single source [47]. When minima and maxima were offered, the average was calculated as the midpoint unless mean group size was explicitly stated, in which case it was used instead (as for primates). Species explicitly described as "solitary" were assigned a mean group size of 1; species for which the "basic social unit" was described as a breeding pair were assigned a mean group size of 2. A mother and her litter were not considered a social group for this analysis, although doing so did not affect the significance of the calculated difference between our primates versus carnivorans. Data for *Zalophus californianus* were not included in [47] and were instead obtained from Table 2 in [64].

## Vocalization data

Vocalization data for all 55 of the species examined here were obtained from [9]. In that study, a mean F0 value for each species was derived from a set of six vocalizations systematically selected to cover the range of spectral variability present in a larger set of vocalizations. This procedure ensured that the vocalization data chosen to represent each species were derived in the same manner across species, promoting valid interspecific assessments. The resulting "F06" value was only available for 80% ($n = 44$) of the species examined here. However, using the same data set and reducing the number of vocalizations required to calculate mean values from six to three, a comparable "F03" value was derived for 96% ($n = 53$) of our species. For species with both F03 and F06 values, these variables were highly correlated (Spearman $r = 0.97$ for primates [$n = 20$] and 0.98 for carnivorans [$n = 24$], $Ps < 0.0001$), justifying the use of F03 values here. For the two species without F03 values (*Lemur catta* and *Otocyon megalotis*), the data in [9] included less than three vocalizations with measurable F0 values. These species were thus excluded from analyses involving F0. It should also be noted that, although several previous studies have preferred using acoustic measures other than mean F0 in their comparisons with body size (e.g., minimum F0, maximum F0, or F0 range; [65,66]), attempting such comparisons across all 53 species included in the present acoustic analysis invariably

resulted in weaker relationships with larynx size than that observed for mean F0: the $R^2$ values for linear models of larynx-size versus log10-transformed F0 mean, maximum, minimum, and range were 0.583, 0.513, 0.506, and 0.361 respectively. Finally, we note that the acoustic allometries reported using F0 mean in [9] are among the strongest in the literature (in the sense of highest $R^2$; cf. [65–67]), providing further support for the systematic approach taken to deriving F0 mean in [9] and, by extension, here.

## Supporting information

**S1 Text. Full description of the body mass results.**
(DOCX)

**S1 Data.** (A) Laryngeal measurements. All data reported in millimeters. (B) Body length, larynx size, covariates, and fundamental frequency. Dashes indicate missing data. Maximum life span data compiled from [59]. Average male and female mass data are compiled from [46,47,68–78]. Relative age = specimen age/maximum life span. Species sexual dimorphism = log10(average male mass)/log10(average female mass). (C) Social group size data.
(XLSX)

**S1 Fig. A reproduction of Fig 4C with abbreviated species names labeling the individual data points.** See Fig 3 for full species names. The data used to create this figure are located in S1 Data, sheet B, columns C and K.
(DOCX)

**S2 Fig. The body length and sex of each selected primate specimen (*n* = 26; top panel) and each selected carnivoran specimen (*n* = 29; bottom panel).** Each circle represents one specimen and is labeled with species name (blue = male, red = female, gray = sex unavailable). Horizontal bars represent body length ranges and vertical bars represent body length means for each species as reported in [46,47]. These ranges and means are given for males (blue) and females (red) separately, when reported separately in [46,47] or in gray when reported together. The four smallest primate specimens and two smallest carnivoran specimens were scanned using micro-CT. The data used to create this figure are located in S1 Data, sheet B, columns B, and G. CT, computed tomography.
(DOCX)

## Acknowledgments

The authors are grateful to Nadja Kavcik-Graumann for drawings in Fig 3B and to Thomas O'Mahoney for comments on an earlier draft. National Museums Scotland thanks the Negaunee Foundation for its generous support of a curatorial preparator, who dissected out the larynges used in this study.

## Author Contributions

**Conceptualization:** Daniel L. Bowling, W. Tecumseh Fitch.

**Data curation:** Daniel L. Bowling, Asha Sato, Georg Hantke, Andrew C. Kitchener.

**Formal analysis:** Daniel L. Bowling, Jacob C. Dunn, Jeroen B. Smaers.

**Funding acquisition:** Andrew C. Kitchener, Michaela Gumpenberger, W. Tecumseh Fitch.

**Investigation:** Daniel L. Bowling, Jacob C. Dunn, Maxime Garcia, Asha Sato, Georg Hantke, Stephan Handschuh, Sabine Dengg, Max Kerney, Andrew C. Kitchener, Michaela Gumpenberger, W. Tecumseh Fitch.

**Methodology:** Daniel L. Bowling, Jacob C. Dunn, Jeroen B. Smaers, Maxime Garcia.

**Project administration:** Daniel L. Bowling.

**Resources:** Andrew C. Kitchener, Michaela Gumpenberger, W. Tecumseh Fitch.

**Supervision:** Daniel L. Bowling, Jacob C. Dunn, W. Tecumseh Fitch.

**Visualization:** Daniel L. Bowling, Jeroen B. Smaers.

**Writing – original draft:** Daniel L. Bowling, Jacob C. Dunn, Jeroen B. Smaers, W. Tecumseh Fitch.

**Writing – review & editing:** Daniel L. Bowling, Jacob C. Dunn, Jeroen B. Smaers, Maxime Garcia, Asha Sato, Georg Hantke, Stephan Handschuh, Andrew C. Kitchener, Michaela Gumpenberger, W. Tecumseh Fitch.

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
