## [Editor Report · Decision Letter 0]

21 Feb 2020

Dear Dr Bowling, 

Thank you for submitting your manuscript entitled "Rapid evolution of the primate larynx?" for consideration as a Research Article by PLOS Biology.

Your manuscript has now been evaluated by the PLOS Biology editorial staff as well as by an academic editor with relevant expertise and I am writing to let you know that we would like to send your submission out for external peer review.

Please re-submit your manuscript within two working days, i.e. by Feb 25 2020 11:59PM.

Kind regards,

Ines

--

Ines Alvarez-Garcia, PhD

Senior Editor

PLOS Biology

Carlyle House, Carlyle Road

Cambridge, CB4 3DN

+44 1223–442810

---

## [Decision Letter · Decision Letter 1]

14 May 2020

Dear Daniel,

Thank you very much for submitting your revised manuscript "Rapid evolution of the primate larynx?" for consideration as a Research Article by PLOS Biology. Thank you also for your patience as we completed our editorial process, and please accept my apologies for the delay in providing you with our decision. As with all papers reviewed by the journal, yours was evaluated by the PLOS Biology editors as well as by an Academic Editor with relevant expertise and by two independent reviewers - note that one of them is one of the original reviewers from eLife.

Based on the reviews, we will probably accept this manuscript for publication, assuming that you will modify the manuscript to address the remaining points raised by the reviewers.

The reviews are attached below. You will see that the reviewers are very positive and only ask for minor clarifications or extra analyses. After discussing the reports with the academic editor, we do feel that you should address/clarify the remaining issues raised by Reviewer 3. Regarding Reviewer 5’s comments, however, we do not think it is necessary to test the hypotheses raised in the discussion, as it would probably not solve the ‘simplicity’ issues previously raised by the reviewers. Thus, while you should reply to the reviewer regarding this point, we would not make this request a requirement for publication. All the other points should be addressed. Please also make sure to address the data and other policy-related requests noted at the end of this email.

We expect to receive your revised manuscript within two weeks. Your revisions should address the specific points made by each reviewer. In addition to the remaining revisions and before we will be able to formally accept your manuscript and consider it "in press", we also need to ensure that your article conforms to our guidelines. A member of our team will be in touch shortly with a set of requests. As we can't proceed until these requirements are met, your swift response will help prevent delays to publication.

*Copyediting*

*Published Peer Review History*

*Early Version*

*Submitting Your Revision*

Sincerely,

Ines

--

Ines Alvarez-Garcia, PhD

Senior Editor

PLOS Biology

Carlyle House, Carlyle Road

Cambridge, CB4 3DN

+44 1223–442810

Note that we do not require all raw data. Rather, we ask that all individual quantitative observations that underlie the data summarized in the figures and results of your paper. You mention that all mass data are compiled from references 44 and 45, and that is fine, but if you processed the data and used new derivatives for your figures then you will need to supply the numerical data underlying the graphs. Also, I can see that you have provided a data file (S1 data), so if this data has been used to make the figures, please specify which ones.

Data can be made available in one of the following forms:

Fig. 2, Fig. 3A, Fig. 4B, C; Fig. S3 and Fig. S4

Reviewers' comments

Rev. 3: James Higham - please note that this reviewer has waived anonymity

I have read the revised version of the manuscript, which I originally reviewed for eLife.

I think that the manuscript is much improved. In particular, I found the description of the analyses much clearer. The manuscript is now more appropriately caveated as a comparison of two taxa. I think the manuscript and data presented are interesting, and it was clearly a lot of work, for which I congratulate the authors.

A couple of remaining comments:

1) Now that I can more clearly understand the analyses that were undertaken, I agree with the authors that the analyses are appropriate as they are. However, I'm confused by the authors' response to my comments about log-transformation of variables. In their response, the authors write: "There are many reasons to log-transform data that extend beyond issues of normality." I certainly agree with this. However, I was not the one who suggested that the data were log-transformed to achieve normal distributions of the variables - this is what the authors themselves use as their justification in their manuscript. In the present version of the MS, they write: "Log-transformations (base-10) were necessary to achieve normality (as indicated by Shapiro-Wilk tests) for all laryngeal measures and body length." (Lines 444-446). If the authors have other reasons for log-transforming all their data, then they should delete this and give a different justification. Otherwise, I don't see that they have addressed my original comment.

2) The caveats to the two-taxon comparison have improved the manuscript substantively. The authors write: "we cannot be certain whether the primate larynx is exceptionally large and variable or the carnivoran larynx exceptionally small and constrained." (Lines 308-309) The authors should change the word "exceptionally" to "relatively". This is what they have shown - that primates are relatively large and more variable compared to carnivores, and vice versa. For all we know both could be relatively invariate compared to many other mammalian orders, or both could be relatively variable compared to many other mammalian orders. In addition to changing 'exceptionally' for 'relatively', I also think that this latter point should be made explicit at this point in the manuscript.

I hope that my additional comments are helpful - James Higham

Rev. 5:

In the same way as all four former reviewers from eLife, I believe that this paper is very interesting and would constitute a very valuable addition to the literature. The data combined in these analyses are very impressive (including 26 primates and 29 carnivorans, collected over 10 years) and the results relatively straightforward. There are also some obvious limitations, as for any analysis that relies on data available online or published in books or by other authors (e.g. exact body size and F0 of the various species included here), in addition to the comparison of 'only' two orders (primates and carnivores). However, I believe that the results presented in this paper deserve to be published, since although the best estimates of the location of the regression lines shown might be revised later on when more exact data about each species is gathered, the results are fairly robust and their publication have the potential to stimulate research in this field.

Since the former reviewers already raised many concerns and I think that the authors responded to these comments very well, I only have a few suggestions to improve the analyses and make the claims stronger.

L80-86. I know that the effects of these covariates is not significant in terms of p-values. However, they might still explain some variance in the data, especially for those that show lower p values (e.g. SS). Since potential systematic differences between primates and carnivorans in the sex and age of your individual larynx specimens is one of the major limitation of the study (also highlighted by former reviewers), would it not make sense to run your pANCOVA on the residual variance of larynx size against body length and these covariates (at least SS)?

210-211. 'relaxation of selective forces related to diet'. Can you give more details here about which kinds of diet can lead to a relaxation of selective forces?

L205-257. Several of the former reviewers highlight the 'simplicity' of the papers and its results, which are also 'only' based on two mammalian orders. I am thus wondering if the paper could not be substantially improved by actually testing these hypotheses mentioned in the discussion (diet, locomotor behaviour (terrestrial versus arboreal), and habitat (in relation to vocalisations))? Are data available on the diet and ecology of all of these species (or a subset at least) not available? If yes, the authors could maybe test which of these hypotheses explain the variation observed in larynx size the best. If this is not feasible, a general conclusion with the opinion of the authors concerning which of these hypotheses is more likely would be appreciated.

L296-298. Please provide the statistical results related to these claims.

---

## [Editor Report · Decision Letter 2]

8 Jul 2020

Dear Dr Bowling,

On behalf of my colleagues and the Academic Editor, Simon William Townsend, I am pleased to inform you that we will be delighted to publish your Research Article in PLOS Biology. 

Early Version

PRESS 

Kind regards,

Alice Musson

Publishing Editor, 

PLOS Biology

on behalf of

Ines Alvarez-Garcia,

Senior Editor

PLOS Biology